# Effects of NaCl Stress on the Growth, Physiological Characteristics and Anatomical Structures of *Populus talassica* × *Populus euphratica* Seedlings

**DOI:** 10.3390/plants11223025

**Published:** 2022-11-09

**Authors:** Ying Liu, Mengxu Su, Zhanjiang Han

**Affiliations:** 1College of Life Science and Technology, Tarim University, Alar 843300, China; 2Xinjiang Production and Construction Corps Key Laboratory of Protection and Utilization of Biological Resources in Tarim Basin, Alar 843300, China

**Keywords:** *Populus talassica* × *Populus euphratica*, NaCl-stress treatments, growth index, physiological characteristics, anatomical structures

## Abstract

In order to elucidate the salt tolerance mechanism of *Populus talassica* × *Populus euphratica*, the growth, physiology and anatomical characteristics of *P. talassica* × *P. euphratica* were studied under different concentrations of NaCl-stress treatments. In this study, the annual seedlings of *Populus talassica* × *Populus euphratica* were used as the test material in a field potted control experiment. The basic salt content of the culture soil was the control (CK), and two NaCl treatments of 200 mmol/L and 400 mmol/L were established. The pot experiment showed that: (1) Compared with CK, the 200 mmol/L NaCl-stress treatment significantly increased the growth parameters of *P. talassica* × *P. euphratica*, such as leaf area, plant height, ground diameter, biomass, root length, root surface area, root fork number and root-shoot ratio. However, compared with CK, the 400 mmol/L NaCl-stress treatment significantly reduced most growth parameters. (2) The 200 and 400 mmol/L NaCl-stress treatments significantly decreased various physiological parameters such as relative water content (RWC), chlorophyll content, water potential, stomatal opening and photosynthetic parameters and increased the accumulation of MDA and Pro compared with CK. The 200 mmol/L NaCl-stress treatment significantly increased the activity of antioxidant enzymes, and the 400 mmol/L NaCl-stress treatment significantly decreased the activity of antioxidant enzymes. (3) Compared with CK, 200 and 400 mmol/L NaCl-stress treatments significantly improved the leaf palisade tissue thickness and palisade-to-sea ratio, as well as the stem xylem and stem phloem thickness and pith diameter, and significantly increased the root xylem thickness, root phloem thickness, and root cross-cutting diameter of *P. talassica* × *P. euphratica*. The growth, physiological characteristics and anatomical characteristics of *P. talassica* × *P. euphratica* under NaCl-stress treatments showed that it had good salt tolerance and adaptability, and the 200 mmol/L NaCl-stress treatment promoted the growth of *P. talassica* × *P. euphratica* to a certain extent. This study provided a theoretical basis for the study of the salt-tolerant mechanism of *P. talassica* × *P. euphratica*.

## 1. Introduction

Soil salinization seriously affects China’s agricultural production and ecological environment [1]. Xinjiang is located in an arid and semi-arid area with little precipitation, abundant sunshine and strong evaporation. Soil salinization is particularly evident in Xinjiang. The total area of salinized soil in Xinjiang is approximately 14.6 million hectares, accounting for 40.7% of the national salinized soil area [2,3]. It is the largest distribution area of salinized soil in China. Saline soils, containing predominantly Na^+^ and Cl^−^, are widespread globally and affect major crop production [4]. Salinity in soil or water represents a significant abiotic stress that alters multiple processes in plants [5]. Abiotic stresses negatively affect the physiology and biochemistry of plants, consequently altering plant growth and development [6].

Depending on the ability of plants to grow in saline environments, they are classified as either glycophytes or euhalophytes and their response to salt stress differs [7,8]. Halophytes, naturally grow in, or even depend on, elevated or high NaCl environments. They can survive and complete their life cycles in media containing more than 200 mmol/L NaCl [9,10,11,12,13,14,15]. Halophytes are remarkable in their abilities to regulate their physiology to adapt to changes in saline conditions [16]. However, glycophytes are defined as species that cannot survive in a saline environment [17]. Glycophytes, which include most crops, cannot grow in the presence of high salt levels; their growth is inhibited or even completely prevented by NaCl concentrations of 100–200 mmol/L, resulting in plant death [18].

Nevertheless, some studies on the salt tolerance of poplar species have used young rooted cuttings for experimental material [19,20,21,22,23,24,25,26]. The choice of younger plants is required by the fast growth rate of most poplars: trees more than 1 or 2 years old become too large for easy handling in the laboratory or greenhouse [27]. *P. talassica* has been widely considered as a model species for elucidating abiotic resistance mechanisms of trees, e.g., responses to salinity or drought stress [28,29,30].

The genus *Populus* is a member of the Salicaceae family and dominant tree species in arid areas, adapted to a variety of conditions, resulting in a rich source of variation in tree morphology, anatomy, physiology and respond to abiotic and biotic stress [31]. *Populus talassica* × *Populus euphratica* is a new cultivar obtained by cross breeding *P. euphratica* as the male parent and *P. talassica* as the female parent. The seedling population has excellent characteristics, such as drought resistance and saline-alkali tolerance, and a fast growth rate. Compared with *P. euphratica*, its asexual reproductive ability is significantly improved. It has certain promotive and planting values in saline-alkali areas with dry and barren soils [32].

The adverse effects of salinity on *Populus* have been investigated in a wide variety of economically important species and ecotypes [33]. The tolerance mechanisms of plants to salinity have been intensively studied in the last decades [34,35,36,37,38]. Plant responses to salinity include a complex set of traits that involve morphological, physiological and cellular processes [39]. However, the current research on the growth, physiology and anatomical characteristics of *P. talassica* × *P. euphratica* is not sufficient. Therefore, there is an urgent need to fully understand the response mechanism of *P. talassica* × *P. euphratica* to salt stress and apply relevant knowledge to improve the salt tolerance of *P. talassica* × *P. euphratica*.

In this experiment, annual potted *P. talassica* × *P. euphratica* seedlings served as experimental materials, and the phenotypic and physiological responses of *P. talassica* × *P. euphratica* to NaCl-stress were studied using the potted method under controlled conditions. Three NaCl concentrations were chosen for treatments in this experiment, being control (CK), 200 and 400 mmol/L NaCl. Growth, physiological characteristics and anatomical structures of *P. talassica* × *P. euphratica* were investigated under the three treatment conditions in this study. The adaptive mechanisms of *P. talassica* × *P. euphratica* to NaCl-stress were explored. The purpose of this study was to provide reliable growth, physiological and structural indicators for the identification of the salt-tolerance mechanisms of *P. talassica* × *P. euphratica* germplasm materials.

## 2. Results

### 2.1. Effects of Salinity on P. talassica × P. euphratica Growth

#### 2.1.1. Effects of Different Concentrations of NaCl-Stress Treatments on Leaf Length, Leaf Width, Leaf Area, Leaf Number and the Survival Rate of *P. talassica* × *P. euphratica*

After 45 d of NaCl-stress treatments, the results showed that under the conditions of 200 mmol/L NaCl-stress treatment, leaf length, leaf width and leaf area of the *P. talassica* × *P. euphratica* seedlings all reached maximum values compared with CK, and the number of leaves increased. When the NaCl concentration increased to 400 mmol/L, the growth indexes of the *P. talassica* × *P. euphratica* seedlings decreased significantly compared with CK, and individual plant deaths occurred, decreasing the survival rate (Table 1).

#### 2.1.2. Effects of Different Concentrations of NaCl-Stress Treatments on *P. talassica* × *P. euphratica* Biomass

After 45 d of NaCl-stress treatments, the results showed that under the conditions of 200 mmol/L NaCl-stress treatment, the aboveground, underground and whole plant biomasses (dry weight) of the *P. talassica* × *P. euphratica* seedlings all reached their maximum values compared with CK. When the NaCl increased to 400 mmol/L, the biomass decreased significantly. In addition, as the NaCl concentration gradually increased, the root/shoot ratio increased (Table 2).

#### 2.1.3. Effects of Different Concentrations of NaCl-Stress Treatments on *P. talassica* × *P. euphratica* Root Vitality

After 45 d of treatments with NaCl-stress, as shown in Table 3, the root length, root surface area, volume, average diameter and number of forks all reached maximum values under the 200 mmol/L NaCl-stress treatment conditions compared with CK. When the NaCl concentration was increased to 400 mmol/L, the root correlation index of *P. talassica* × *P. euphratica* was significantly decreased and growth was inhibited (Table 3).

#### 2.1.4. Effects of Different Concentrations of NaCl-Stress Treatments on the Plant Height, Ground Diameter and Crown Width of *P. talassica* × *P. euphratica*

The measured values are shown in Table 4. After 45 d of NaCl-stress treatments, there were no significant differences in growth indexes of the *P. talassica* × *P. euphratica* seedlings in the control group and the treatment group before the NaCl-stress treatments. The plant height, ground diameter, crown width and relative growth reached maximum values under the 200 mmol/L NaCl-stress treatment conditions compared with CK. The growth indexes decreased significantly under the 400 mmol/L NaCl-stress treatment (Table 4).

### 2.2. Effects of Different Concentrations of NaCl-Stress Treatments on P. talassica × P. euphratica Physiological Properties

#### 2.2.1. Effects of Different Concentrations of NaCl-Stress Treatments on Leaf Surface Salt Secretion of *P. talassica* × *P. euphratica*

After 45 d of NaCl-stress treatments, the leaf surface of *P. talassica* × *P. euphratica* is observed with a microscope. It was found that the *P. talassica* × *P. euphratica* leaves have microhairs on their adaxial surface furrows that secrete salt. As shown in Figure 1, compared with CK, under the 200 and 400 mmol/L NaCl-stress treatments, the main veins of *P. talassica* × *P. euphratica* leaves and the adjacent leaf surfaces secreted large volumes of salt crystals, and more salt crystals were secreted on the leaf surface the 400 mmol/L NaCl-stress treatment group (Figure 1).

#### 2.2.2. Effects of Different Concentrations of NaCl-Stress Treatments on the Diurnal Variation in the *P. talassica* × *P. euphratica* Leaf Water Potential

After 45 d of NaCl-stress treatments, the leaf water potential of *P. talassica* × *P. euphratica* was determined, and the diurnal variation curve of water potential was shown in Figure 2. From 8:00 to 20:00, the leaf water potential showed regular changes. The variation trends in the three treatments were basically the same, with the water potential being low at 8:00 in the morning and increased at 11:00 a.m. The water potential reached its lowest value at 14:00, then gradually increased, reaching a maximum value at 17:00, and then, it gradually decreased. As the NaCl concentration increased, compared with CK, the water potential of *P. talassica* × *P. euphratica* leaves gradually decreased, reaching its lowest value of approximately −3.80 under the 400 mmol/L NaCl-stress treatment. The water potential of the CK group was the largest, with a maximum value of approximately −0.75 (Figure 2).

#### 2.2.3. Effects of Different Concentrations of NaCl-Stress Treatments on the Relative Water Contents and Water Saturation Deficits of *P. talassica* × *P. euphratica* Leaves

After 45 d of NaCl-stress treatments, the RWC of the leaves of *P. talassica* × *P. euphratica* was determined and the WSD was calculated. The RWC gradually decreased as the NaCl concentration increased (Figure 3). Compared with CK, when the NaCl concentration reached 400 mmol/L, the RWC of the leaves decreased sharply. The differences between the RWC of CK and those that were treated were more significant. During the treatment period, the WSD was between 14.0% and 19.4%. The higher the NaCl concentration, the more serious the leaf water loss, which further decreased the RWC of the leaves (Figure 3).

#### 2.2.4. Effects of Different Concentrations of NaCl-Stress Treatments on the *P. talassica* × *P. euphratica* Leaf Chlorophyll Content

As shown in Table 5, compared with CK, as the NaCl concentration increased, the chlorophyll content gradually decreased. The chlorophyll content of the 400 mmol/L NaCl treatment group was significantly lower than that of the CK group, and the chlorophyll a/b also significantly decreased (Table 5). The total chlorophyll content of the 400 mmol/L NaCl-stress treatment group was less than 50% that of the CK, and the difference was significant (*p* < 0.05). At the higher NaCl concentration, the chlorosis of plant leaves was more severe, which further led to a decrease in leaf chlorophyll content.

#### 2.2.5. Effects of Different Concentrations of NaCl-Stress Treatments on the Size, and the Opening and Closing, of *P. talassica* × *P. euphratica* Leaf Stomata

After 45 d of NaCl-stress treatments, the changes in *P. talassica* × *P. euphratica* leaf stomata in the three groups, CK, 200 mmol/L NaCl and 400 mmol/L NaCl, were observed using a microscope. Figure 4A–C show the leaf stomatal morphology of the three treatment groups, CK, 200 mmol/L NaCl and 400 mmol/L NaCl, respectively, under a microscope at a 10× magnification. As shown in Figure 4A–C, the number of stomata in the CK group leaves was relatively high, whereas the number of stomata in the 400 mmol/L NaCl-stress treatment group leaves decreased sharply. Figure 4D–F shows the leaf stomatal morphology of the CK, 200 mmol/L NaCl and 400 mmol/L NaCl-stress treatment groups, respectively, at a 40× magnification. As shown in Figure 4D–F, the proportion of open stomata in the CK group leaves was larger, and the proportions of open stomata in the 200 and 400 mmol/L NaCl-stress treatment groups leaves decreased sharply compared with the CK group.

After 45 d of NaCl-stress treatments, the stomata of the leaves of *P. talassica* × *P. euphratica* were observed and measured, and the effects of different concentrations of NaCl on the size, and the opening and closing, of *P. talassica* × *P. euphratica* leaf stomata are shown in Table 6. Compared with CK, the stomatal lengths in the 200 mmol/L NaCl-stress treatment group leaves did not change significantly, whereas the stomatal length in the 400 mmol/L NaCl-stress treatment group leaves significantly decreased. As the NaCl concentration increased, the stomatal width, stomatal area, stomatal density, stomatal area index and the proportion of open stomata in *P. talassica* × *P. euphratica* leaves significantly decreased. Different NaCl-stress treatments closed the stomata to different degrees, and they significantly affected the number of stomata, which were significantly different from the number in the CK group. This indicated that plants could rapidly self-regulate, make adaptive changes, and show increased salt resistance under a certain level of NaCl-stress.

#### 2.2.6. Effects of Different Concentrations of NaCl-Stress Treatments on the Photosynthetic Characteristics of *P. talassica* × *P. euphratica* Leaves

After 45 d of NaCl-stress treatments, the photosynthetic index of the leaves of *P. talassica* × *P. euphratica* was determined, and the effects of different concentrations of NaCl treatment on the photosynthetic characteristics of *P. talassica* × *P. euphratica* were shown in Figure 5. The effects of NaCl-stress on the Pn, Gs, Tr, and Ls values of *P. talassica* × *P. euphratica* seedlings were significantly different (*p* < 0.05). Compared with CK, 200 and 400 mmol/L NaCl-stress treatments significantly reduced Pn, Gs, Tr and Ls values in *P. talassica* × *P. euphratica* leaves, but increased Ci values significantly. NaCl-stress treatments had no significant effect on the WUE value of *P. talassica* × *P. euphratica*, but the average value was slightly greater than that of the CK, indicating an upward trend (Figure 5).

#### 2.2.7. Effects of Different Concentrations of NaCl-Stress Treatments on the Contents of MDA, Pro, Soluble Sugar and Soluble Protein in *P. talassica* × *P. euphratica* Leaves

After 45 d of NaCl-stress treatments, the content of MDA, Pro, soluble sugar and soluble protein content in the leaves of *P. talassica* × *P. euphratica* under different concentrations of NaCl-stress treatments was determined, and the measurement results were shown in Figure 6. As shown in Figure 6A, compared with CK, the 200 and 400 mmol/L NaCl-stress treatments significantly increased the MDA content and reached a maximum at 400 mmol/L NaCl-stress treatment, and the difference was extremely significant (*p* < 0.05). As shown in Figure 6B, compared with CK, the 200 and 400 mmol/L NaCl-stress treatments significantly increased the Pro content and reached a maximum at 400 mmol/L NaCl-stress treatment, and the difference was extremely significant (*p* < 0.05).

As shown in Figure 6C, when the NaCl concentration reached 200 mmol/L, the soluble sugar content decreased significantly compared with CK, and when the NaCl concentration reached 400 mmol/L, the soluble sugar content reached its highest value (Figure 6C). As shown in Figure 6D, compared with CK, when the NaCl concentration was 200 mmol/L, the soluble protein content decreased significantly compared with CK, and when the NaCl concentration reached 400 mmol/L, the soluble protein content reached maximum values, which was greater than that of the CK (Figure 6D).

#### 2.2.8. Effects of Different Concentrations of NaCl-Stress Treatments on the Activities of Three Antioxidant Enzymes and the Relative Conductivity in *P. talassica* × *P. euphratica* Leaves

As shown in Figure 7A, compared with CK, when the NaCl concentration was at 200 mmol/L, the SOD activity reached maximum values. When the concentration of NaCl increased to 400 mmol/L, the SOD activity decreased significantly. As shown in Figure 7B, compared with the CK group, at 200 mmol/L, the POD activity was highest. When the NaCl concentration increased to 400 mmol/L, the POD activity decreased significantly. As shown in Figure 7C, compared with CK, when the NaCl concentration was 200 mmol/L, the CAT activity reached its maximum. When the NaCl concentration increased to 400 mmol/L, the CAT activity decreased significantly. Conductivity is a measure of the ease with which electrical current will pass through water; the greater the salinity, the greater the conductivity [37]. The relative conductivity measurements are shown in Figure 7D. Compared with CK, the 200 and 400 mmol/L NaCl-stress treatments significantly increased the relative conductivity content of the blades, and when the NaCl concentration was 400 mmol/L, the relative electrical conductivity of leaves reached maximum values. 

### 2.3. Effects of Different Concentrations of NaCl-Stress Treatments on the Root, Stem and Leaf Anatomy of P. talassica × P. euphratica

#### 2.3.1. Effects of Different Concentrations of NaCl-Stress Treatments on the Leaf Anatomy of *P. talassica* × *P. euphratica*

The *P. talassica* × *P. euphratica* mesophyll cells were composed of upper and lower epidermal cuticle, upper and lower epidermis, palisade tissue, spongy tissue, xylem, phloem, cambium, parenchyma and stomata (Figure 8).

The leaf vascular bundle structure of *P. talassica* × *P. euphratica* was well developed, and it was approximately oval, consisting of xylem, phloem and the vascular cambium between the xylem and phloem. The xylem and phloem were connected by a vascular cambium (Figure 9A–C). The palisade tissue was composed of two to three layers of parenchymal cells, which were closely arranged, and the cells were long cylindrical in shape and contained many chloroplasts (Figure 9D–F). The spongy tissue was irregular in shape, loosely arranged and had obvious intercellular spaces between the tissues. It was composed of five to six layers of loosely arranged cells of approximately circular size, with more crystals distributed (Figure 9G–I).

The effects of NaCl-stress on leaves were mainly reflected in the thickness of palisade and spongy tissues. As shown in Table 7, compared with CK, the thickness of palisade tissues in 200 mmol/L and 400 mmol/L NaCl-stress treatment groups increased by 33.30% and 64.62%, respectively, the thickness of sponge tissue in 200 mmol/L and 400 mmol/L NaCl-stress treatment groups decreased by 10.84% and 24.43%, respectively, and the differences were significant (*p* < 0.05). Moreover, the 200 and 400 mmo/L NaCl-stress treatments significantly increased the leaf thickness, CP and the tightness of the blade structure of *P. talassica* × *P. euphratica*.

#### 2.3.2. Effects of Different Concentrations of NaCl-Stress Treatments on the Stem Anatomy of *P. talassica* × *P. euphratica*

The cross-section of a *P. talassica* × *P. euphratica* stem is approximately circular, consisting of epidermis, pericyte, cortical parenchyma, xylem, phloem and pith in the center (Figure 10).

The stem vascular cambium was obvious, the xylem was arranged radially and the number of xylem vessels was large (Figure 11A–C). The phloem was relatively narrow and contained bast fibers inside (Figure 11D–F). The central medulla was well developed and consisted of large water-storing parenchymal cells, and there were discrete individual crystal-containing cells in the medulla and medullary rays (Figure 11G–I). The phloem contained many phloem fibers, which were embedded in the phloem and consisted of many granular cells (Figure 11J–L).

The measurements of stem anatomy are shown in Table 8. As shown in Table 8, the 200 and 400 mmol/L NaCl-stress treatments significantly increased the thickness of the epidermis and cortex of stem, the thickness of the phloem and xylem and the diameter of the pith. Under the treatment of 400 mmol/L NaCl-stress, the stem indicators reached maximum values, and the differences were significant (*p* < 0.05).

#### 2.3.3. Effects of Different Concentrations of NaCl-Stress Treatments on the Root Anatomy of *P. talassica* × *P. euphratica*

The secondary structures of *P. talassica* × *P. euphratica* roots, from outside to inside, consisted of pericytes, secondary phloem, vascular cambium and secondary xylem (Figure 12).

Due to the thickening of the root, the epidermis and cortex break, and the cork cambium formed a new protective tissue, the pericyte. The pericyte cells were oblong, neatly and tightly arranged and shed (Figure 13A–C). The xylem contained a large number of vessels that transport water, and wood rays were also arranged in the xylem (Figure 13D–F). The secondary phloem was located in the cortex and was composed of phloem parenchymal cells. The cells were irregular in shape and tightly arranged. There was a large number of storage cells and secretory cells distributed among the phloem cells (Figure 13G–I). The phloem was also inlaid with a small amount of bast fibers. The bast fibers were composed of many round cake-like particles embedded in the phloem, and all the bast fibers together formed a ring (Figure 13J–L).

Under NaCl-stress, plant roots adopted their own adaptive strategies. The measurements of root secondary structures are shown in Table 9. Compared with CK, the 200 and 400 mmol/L NaCl-stress treatments significantly increased the root xylem thickness, phloem thickness, xylem vessel diameter and root cross section diameter of *P. talassica* × *P. euphratica*. All of the above indicators reached maximum values under 400 mmol/L NaCl-stress treatment (Table 9).

## 3. Materials and Methods

### 3.1. Overview of the Study Area

The study area is located in the seedlings base of the 10th Regiment of the 1st Division of the Xinjiang Production and Construction Corps (40°34′6.14″ N; 81°15′32.64″ E and altitude of 1014 m). It belongs to an extreme continental arid desert climate in the warm temperate zone. The extreme high temperature reaches 35 °C, the extreme minimum temperature is −28 °C and the annual average sunshine is 2556.3–2991.8 h. There is very little rainfall, little snow in winter and strong surface evaporation. The average annual precipitation is 40.1–82.55 mm, and the average annual evaporation is 1876.6–2558.9 mm. The soil type is mainly sandy.

### 3.2. Plant Material and Imposition of Salt Stress

Annual potted *P. talassica* × *P. euphratica* seedlings served as experimental materials, and the seedlings were treated with different concentrations of NaCl using the open-air potted soil cultivation method. The potted seedling had plant heights of 61–81 cm, ground diameters of 0.8–1.2 mm and crown widths of 16–34 cm. The 16 cm high plastic pot had an upper diameter of 24 cm and bottom diameter of 12 cm. The pots contained 9.82 kg of nursery mellow soil: pH 7.9; soil salinity, 0.49 g/kg; soil density, 1.36 g/cm^3^; soil conductivity, 852.6 µs/cm; soil organic matter content, 21.04 g/kg; and available potassium content, 148.96 mg/kg. The contents of available phosphorus and alkali-hydrolyzed nitrogen were 7.49 mg/kg and 139.58 mg/kg, respectively.

The experiment was conducted in a completely randomized design. The selected 1-year-old potted *P. talassica* × *P. euphratica* seedlings showed consistent growth, good growth and no pests and diseases. They were divided into three groups, with 20 plants in each group, totaling 60 plants that were then subjected to NaCl-stress treatments. The salt content of the soil itself (4.8 g/pot; CK) was used as a control, and two other concentration gradients were set as 200 mmol/L and 400 mmol/L. To avoid salinity shock and to acclimatize seedlings to high NaCl concentrations, *P. talassica* × *P. euphratica* was treated with increasing NaCl concentrations progressively until the predetermined concentrations were reached, and the NaCl concentration in the basin was kept constant [40]. The 200 and 400 mmol/L NaCl-stress treatment groups were watered with 1 L of the prepared NaCl solution per pot each time, and the CK group was irrigated with 1 L of deionized water per pot each time. Treatments were performed every 3 d for a total of five treatments, eventually reaching the preset NaCl concentration. After maintaining the same growth conditions for 45 d, the growth, physiological indicators, photosynthetic parameters and anatomical structures of roots, stems and leaves were observed and measured.

### 3.3. Growth Index Measurement Methods

Growth indexes were measured before and after NaCl-stress treatments for 45 d. Plant height was measured using a tape measure to an accuracy of 0.1 cm [41]. The ground diameter was measured with an electronic digital Vernier caliper with an accuracy of 0.02 mm [25]. The plant heights, ground diameters and crown widths of seedlings in the three treatment groups before and after 45 d of NaCl-stress treatments were counted, as were any differences.

After 45 d of NaCl-stress treatments, 10 leaves with good growth and no disease and insect pests were taken from each plant in the CK group and the treatment groups. Using a leaf area meter to acquire data such as leaf length, leaf width, and leaf area. Simultaneously, the number of leaves per seedling in the three treatment groups was counted, and the final calculations were averaged [27].

Five seedlings of *P. talassica* × *P. euphratica* were selected from the treatment group and the CK group, totaling 15 seedlings, and they were dug out with the root systems. After excavating whole plants, they were taken to the laboratory and washed with deionized water. The above- and the underground parts were separated, placed in labeled envelopes, fixed in an oven at 110 °C for 20 min and then placed in an oven at 80 °C to dry to constant weights. The dry masses of the above- and the underground parts of each plant were weighed [26]. The sum of the dry masses of the two parts of each plant represented the dry mass of the whole plant. The ratio of the dry mass of the underground part (dry mass of roots) to the dry mass of the aerial part (dry mass of stems and leaves) of each plant represented the root/shoot ratio.

Additionally, whole *P. talassica* × *P. euphratica* seedlings were taken out of the flowerpots, rinsed with deionized water and scanned using a Microtek i800 Plus scanner to obtain root images. Then, the root system configuration parameters, such as root length, root surface area, root volume, root average diameter and number of forks, were measured using an LA-S series plant image analyzer system. The measurements of all the growth indicators were repeated three times, and the statistical results were averaged.

### 3.4. Physiological Index Measurement Methods

The salt secretion on the leaf surfaces of *P. talassica* × *P. euphratica* in different NaCl-stress treatment groups were observed and imaged using a microscope (Nikon SMZ1500), and the salt crystals secreted on the leaf main vein and the adjacent leaf surfaces were observed and imaged. After 45 d of NaCl-stress treatments, five plants were taken from each treatment group, three leaves were cut from the same part of each seedling and the chlorophyll contents of the leaves were determined by ethanol extraction [41]. At the same time, three plants were taken from each treatment group and five leaves were cut from the same part of each seedling. The relative water contents (RWC) of the leaves were measured using the weighing method [41]. First, the fresh weight Wf of the leaves were found, and then the leaves were immersed in distilled water for several hours to make the leaves absorb water into a saturated state. The leaves were taken out and the water absorbed on the surface with absorbent paper, immediately put into a weighing bottle of known weight then put in distilled water for a period of time to absorb the moisture on the outside and finally weighed again until the weight no longer increased. At this time, Wt is the weight when the leaf is saturated with water, after which the sample is dried to obtain the tissue dry weight Wd and the RWC of the leaf is calculated. At the same time, the percentage of WSD is calculated:RWC = (Wf − Wd)/(Wt − Wd) × 100%(1)
WSD = (1 − RWC) × 100%(2)

In total, 10 leaves from each treatment were picked and leaf imprints from the backs of the leaves were lifted using nail polish (three separate regions per leaf). Stomatal density was assessed using an optical Leitz microscope (Leitz DIA LUX 22EB) equipped with a digital camera (Hitachi KP-D 40 Color Digital). Stomata were counted using the analysis software program for image analysis (Delta-T Devices Ltd., Cambridge, UK).

Plant leaves of the same size were selected and washed them with distilled water three times. Filter paper was used to absorb the surface moisture, and the leaves were cut into strips of suitable length (avoiding opening the main vein). For each treatment group, 0.1 g of three fresh samples were weighed, placed into 10 mL deionized water in graduated test tubes, covered with glass stoppers and soaked at room temperature for 12 h. A conductivity meter was used to measure the conductance (R1) of the leaching solution. Samples were then heated in a boiling water bath for 30 min, cooled to room temperature, shaken and the conductance (R2) of the leaching solution was remeasured. Relative conductivity = R1/R2 × 100%.

After a 45 d NaCl-stress treatments, on a sunny day, the leaf water potential was measured every 3 h from 08:00 to 20:00, using a pressure chamber water potential meter (PMS Instrument Company, Albany, OR, USA) to determine the diurnal variation. Three seedlings were measured for each treatment, and three leaves were measured per young tree. The leaves selected were in the same position, with uniform illumination, and showed healthy consistent growth (mature leaf samples were all from in the middles of peripheral branches). The average value of three leaves, with nine replicates, was taken to represent the average leaf water potential of *P. talassica* × *P. euphratica*.

The malondialdehyde (MDA) content was determined using the thiobarbituric acid method. The Pro content was determined by ninhydrin colorimetry [41], and superoxide dismutase (SOD) activity was measured using the nitroblue tetrazolium method [42]. Peroxidase (POD) activity was determined using the guaiacol method [41], and catalase (CAT) activity was determined using the UV absorption method [42]. The soluble sugar content was determined using the anthrone colorimetric method [41], and the soluble protein content was determined using the Coomassie brilliant blue method [43]. Physiological and biochemical indicator kits were purchased from Suzhou Keming Biotechnology Co, Ltd., Suzhou, China. All the physiological index measurements were repeated three times, and the statistical results were averaged.

Photosynthetic indicators were determined using a LI-6400 portable photosynthesis system when the weather was sunny and sky was cloudless. The measurement period was 9:00–11:00. The middle leaves of the plants were used to measure net photosynthetic rate (Pn), transpiration rate (Tr), intercellular CO_2_ concentration (Ci) and stomatal conductance (Gs). The stomatal limitation (Ls) and leaf water-use efficiency (WUE) were calculated. The measured light intensity was 1100 μmol·m^−2^.s^−1^, the temperature was approximately 25 °C, and the atmospheric CO_2_ concentration (Ca) was approximately 400 μmol·mol^−1^. In total, 10 plants were measured per treatment and 3 leaves from the same part of each plant were selected for measurements, which were averaged. The formulae for calculations were as follows:Ls = 1 − Ci/Ca(3)
WUE = Pn/Tr(4)

### 3.5. Determination of the Anatomical Structural Index of Roots, Stems, and Leaves

After 45 d of NaCl-stress treatments, five plants were taken from each treatment group, and the roots, stems and leaves of each *P. talassica* × *P. euphratica* seedling were taken, fixed and preserved in FAA solution. These were transformed into permanent film using the paraffin section method [44]. For leaves, the palisade tissue thickness, spongy tissue thickness and leaf thickness were measured with an industrial digital camera (OPLENIC CORP, Zhejiang, China). For roots, the periderm, phloem and xylem thicknesses, vessel diameter and root diameter were measured under a Leica microscope. For stems, the stem epidermis, cortex, phloem, xylem and pith thickness were measured with an industrial digital camera (OPLENIC CORP) [45]. Five fields of view were observed for each root, stem and leaf section, and 10 structures were observed in each field of view. The average values of the anatomical structural parameters of the rhizomes and leaves from five fields of view were used as the values for *P. talassica* × *P. euphratica*. The CP and tightness of leaf tissue structure were calculated suing the following formulae:CP = Palisade tissue thickness/Spongy tissue thickness(5)
Tightness of leaf tissue structure = (Palisade tissue thickness/Blade thickness) × 100%(6)

### 3.6. Data Processing and Analyses

The experimental data were analyzed, processed and plotted using Microsoft Excel 2010 software. Statistical analyses were performed using SPSS 25.0 software. Significant differences between means were determined using the least significant difference test at *p* < 0.05.

## 4. Discussion

### 4.1. Growth Characteristics of P. talassica × P. euphratica

Plant responses to salinity vary with the degree and duration of stress and the developmental stage at time of exposure [46]. Many halophytes require a quite high concentration of NaCl (100–200 mmol/L) for optimum growth [12]. Saline-alkali stress has serious effects on plant growth rate [4,14], biomass, plant height, leaf area and root length [47,48,49,50]. Traits such as survival and leaf damage have been the most common criteria for identifying salinity tolerance [51,52].

Some reports have showed clearly that salt stress inhibits the growth of poplar [53,54,55]. In this study, compared with CK, the 200 mmol/L NaCl-stress treatment significantly increased the leaf length, leaf width, leaf area and leaf numbers of *P. talassica* × *P. euphratica*, and it promoted the growth of *P. talassica* × *P. euphratica*. The 400 mmol/L NaCl-stress treatment inhibited the growth of *P. talassica* × *P. euphratica* and decreased the survival rate, but it caused no obvious damage to the leaves of *P. talassica* × *P. euphratica*. Under the 400 mmol/L NaCl-stress treatment conditions, the leaves of *P. talassica* × *P. euphratica* became smaller, which decreased the water use by the plant, thereby allowing it to conserve soil moisture and prevent an escalation in the salt concentration [56]. This is similar to a study by Z Zhu et al. (2012), in which the control, moderate NaCl conditions (200 mmol/L NaCl) significantly increased the leaf dry weight, leaf area and leaf numbers of *Bruguiera gymnorrhiza*, whereas the severe NaCl-stress treatment (500 mmol/L NaCl) significantly decreased each growth parameters [39]. In addition, S Chen et al. (2001) observed that leaf abscission of the *P. euphratica* and the hybrid *P. talassica* × (*P. euphratica* + *Salix alba*) tended to increase with increasing NaCl in the soil [23]. This is also similar to our study.

Under 200 mmol/L NaCl-stress treatment, the survival rate of *P. talassica* × *P. euphratica* remained unchanged, while under 400 mmol/L NaCl-stress treatment, the survival rate of *P. talassica* × *P. euphratica* was 91.6%, which was significantly lower than the CK. Similar to our findings, Shin Watanabe et al. (2000) observed the survival ratio of Pyramidalis *× P. tomentosa* was 78% at 150 mmol/L NaCl-stress treatment and 0% at 250 mmol/L NaCl-stress treatment [57]. Moreover, salt stress affects plant growth and development by influencing fresh and dry weights of roots and shoots, as well as shoot length [58]. In general, plant height and fresh and dry biomasses are inhibited by high levels of salt stress [59,60,61]. In this study, the 200 mmol/L NaCl-stress treatment significantly increased the aboveground, belowground, and whole plant biomasses and promoted the dry matter accumulation of *P. talassica* × *P. euphratica*. Furthermore, the 200 mmol/L NaCl-stress treatment significantly increased root length, root surface area, root volume and average root diameter of *P. talassica* × *P. euphratica*, whereas the 400 mmol/L NaCl-stress treatment decreased the biomasses, and the root-related indexes decreased significantly. However, the root/shoot ratio increased gradually along with the NaCl concentration. Similar to our findings, E Bijanzadeh et al. (1997) reported that salt stress increased root/shoot ratios in Afzal and Karoon cultivars [62].

Root development and activity were inhibited when *P. talassica* × *P. euphratica* was grown in a saline environment (such as 400 mmol/L NaCl), which caused a series of depressive effects on the physiological functions of the aerial plant parts [63,64]. This further decreased the dry matter mass of the aboveground parts [63,64]. In addition, here, the reduction in the shoot dry weight of *P. talassica* × *P. euphratica* under the high salt stress (400 mmol/L NaCl) treatment may also be due to the reductions in plant height and leaf number. This is similar to a study by WH Bolu et al. (2004) observed low salinity stimulated root and shoot biomass formation of *Populus canescens*, resulting in increased biomass as compared to controls over the whole experimental period. The reduction in root length and number reflects the sensitivity of this organ to salt stress [55]. In this study, the 200 mmol/L NaCl-stress treatment significantly promoted the growth of the aerial and underground parts of the roots of *P. talassica* × *P. euphratica*, indicating that the cultivar has a certain salt tolerance [65]. The 200 mmol/L NaCl-stress treatment significantly increased the plant height, ground diameter and crown width of *P. talassica* × *P. euphratica*, whereas the 400 mmol/L NaCl-stress treatment significantly decreased these growth parameters. Similar to our findings, N Orcen et al. (2016) reported that a 200 mmol/L NaCl treatment group had the tallest Salicornia plants, whereas the 500 mmol/L NaCl treatment group had the shortest Salicornia plants [42]. Here, the moderate salinity (200 mmol/L NaCl) had a positive effect on the growth of *P. talassica* × *P. euphratica*. In addition, WH Boluet al. (2004) observed immediate inhibition of root length growth of *Populus canescens* after exposure to high salinity [66].

### 4.2. Physiological Characteristics of P. talassica × P. euphratica

The trees allocated large proportions of NaCl into the leaves, which served as a salt elimination mechanism [29]. A large number of halophytes show the presence of specialized salt secretory structures in their leaves that are epidermal in origin and essentially modified trichomes [67,68]. In this study, the main vein of leaves in the NaCl-stress treatment group, and the surfaces of adjacent leaves, secrete large amounts of salt crystals. Compared with the CK, the 400 mmol/L NaCl-stress treatment group secreted more salt crystals on the *P. talassica* × *P. euphratica* leaf surfaces. This is because *P. talassica* × *P. euphratica* transfers excess salt absorbed by the root system to the leaves in response to salt stress [69].

Under salt stress, leaf water physiology also changed adaptively [55]. Water potential is an important indicator for the estimation of the water status of a plant [70,71]. High salt stress disrupts homeostasis in water potential and ion distribution. Drastic changes in ion and water homeostasis lead to molecular damage, growth arrest and even death [72]. In this study, the diurnal variations in the water potential of *P. talassica* × *P. euphratica* at all the observed time points showed the same trend. The treatment with the highest NaCl concentration had the most negative water potential during the day and the lowest water potential at noon. Trends in the control and treatment groups were consistent throughout the experiment. The greater the negative water potential of *P. talassica* × *P. euphratica* seedlings under NaCl-stress, the greater the effect of NaCl-stress on the water supply of *P. talassica* × *P. euphratica* seedlings. The higher the concentration of osmotically active Na^+^ and Cl^−^ ions, the lower the water potential of the soil and the greater the negative water potential of the plant, which is an adaptation that enables the plant to absorb water from the soil [73].

Salinity changes the RWC of the leaves [74]. In this study, compared with the CK, the RWC of the leaves and chlorophyll content of *P. talassica* × *P. euphratica* leaves significantly decreased under different levels of NaCl-stress treatments. This decrease in leaf relative water content (RWC) could be caused by a lower water availability under stress conditions or by root systems that are not able to compensate for water lost by transpiration through a reduction in the absorbing surface [75,76]. In addition, the total chlorophyll concentration significantly decreased after exposing plants to higher salinity, which may result in the destruction of chloroplast structures, eventually leading to a decrease in the chlorophyll content [77]. Likewise, M Gorai et al. (2011) observed that the parameters of all leaf water relations in Phragmites decreased as the NaCl concentration increased [78]. Furthermore, S Cha-Um et al. (2009) also showed that the chlorophyll a and b, total chlorophyll and total carotenoid contents of sugarcane shoots are reduced due to decreases in salt stress, which is also similar to our study results [79].

Studies have confirmed that the inhibitory effect of salt stress on physiological and biochemical processes, which inhibits photosynthesis and destroys cell membranes [80]. Photosynthesis is the most important process for forming organic matter in plants, and the effects of water and salinity on photosynthesis determine the growth and survival of the plants [81]. In this study, compared with the CK, the Pn, Gs, Tr and Ls values of *P. talassica* × *P. euphratica* leaves gradually decreased as the NaCl concentration increased, whereas the Ci value increased gradually. This resulted from the photosynthetic capacity of the mesophyll cells of *P. talassica* × *P. euphratica* seedlings being further reduced, along with the utilization of CO_2_, resulting in excess CO_2_ and a corresponding reduction in photosynthetic products. Similar to our findings, HC Ma et al. (1997) observed that when the high salt (200 mmol/L NaCI) treatments, photosynthesis of both *P. euphratica* and hybrids decreased 1 day after irrigation with a high salinity solution. There was a significant reduction by day 5 for the hybrids and by day 10 for *P. euphrutica* [21]. In general, Pn reduction as response to an initial osmotic shock, caused by moderate drought or salinity stress, resulted from a stomatal closure [82].

The researchers found that salinity caused physiological modifications in plants such as stomatal density, shape and size [83]. NaCl-stress caused different degrees of stomatal closure and decreased stomatal density. Under salt stress, plant leaves close through stomata or reduce leaf stomatal density to reduce the transpiration rate [84]. In this study, the stomatal length, stomatal width, stomatal area, stomatal density and stomatal opening rate of *P. talassica* × *P. euphratica* leaves were decreased to varying degrees by salt stress. Similar to our findings, V. D. Rajput et al. (2015) observed three months of enhanced salinity resulted in increased stomatal density of one year old *P. euphratica* seedlings at 50, 100 and 150 mmol/L NaCl, whereas at 200 mmol/L salt concentration, it decreased in comparison with other salt treatments, and the area of stomata was declined as well. Salinity also affected the length of stomata openings; it decreased in salt-treated plants to reduce evaporation [85]. Improving stomatal regulation has been regarded as the most effective approach to alleviate salt stress in plants [86,87].

In this study, 200 mmol/L and 400 mmol/L NaCl-stress treatments significantly increased Pro and MDA contents in *P. talassica* × *P. euphratica* leaves. In the study of Zhang Min et al., under the 50–200 mmol/L NaCl-stress treatments, compared with the control, the MDA content of *P. talassica* × *P. euphratica* seedlings treated with 50–150 mmol/L NaCl solution had no significant change, and tended to normal physiological indicators. This indicated that *P. talassica* × *P. euphratica* seedlings were weakly affected by NaCl solution at the concentration of 50–150 mmol/L, and when the NaCl concentration reached 200 mmol/L, the MDA content of *P. talassica* × *P. euphratica* seedlings increased significantly [2]. This was because 200 mmol/L NaCl-stress treatment aggravated the lipid peroxidation of *P. talassica* × *P. euphratica* seedlings and damaged the cell membrane.

In addition, the relationship between the concentration of proline and the reaction to salt stress has been confirmed for many species of plant organisms, including poplar and aspen [18]. Similar to our findings, Shin Watanabe et al. (2000) observed that a high proline content in *P. euphratica* grown under both mannitol and NaCl-stress [57]. The soluble sugar and soluble protein contents significantly increased under the 400 mmoll/L NaCl-stress treatment, which may be involved in the salt-stress resistance of *P. talassica* × *P. euphratica*.

In this study, 200 and 400 mmol/L NaCl-stress treatments significantly increased the relative electrical conductivity (REC) of *P. talassica* × *P. euphratica* leaves. In the study of Zhang Min et al., under the 50–200 mmol/L NaCl-stress treatments, the relative conductivity (REC) of leaves of *P. talassica* × *P. euphratica* seedlings did not change significantly. This is because 50–200 mmol/L NaCl-stress treatments did not change the cell membrane permeability of *P. talassica* × *P. euphratica* seedlings [2].

Under normal conditions, ROS production and scavenging is well regulated. The enzymatic ROS scavenging mechanisms in plant included production of superoxide dismutase (SOD), peroxidase (POD), catalase (CAT) and so on, which can minimize the cellular damage caused by ROS [88]. In this study, compared with the CK, the low salt concentration increased the activities of the three antioxidant enzymes, whereas their activities decreased significantly at the high salt concentration. SOD, POD and CAT decreased significantly in plants under higher salinity conditions, suggesting that the excess H_2_O_2_ was not effectively scavenged, thereby causing more serious oxidative stress. This may lead to decreases in the activity levels of the three antioxidant enzymes. Similar to our findings, V. D. Rajput et al. (2015) observed anti-oxidative activity after three months of salt stress in one-year-old *P. euphratica* seedlings, and results showed POD activity increased with increasing rate of salt concentrations at all level of salt treatments (50, 100, 150, 200 mmol/L NaCl) [85]. A high POD activity may play an important defensive role against salt stress [89].

### 4.3. Anatomical Characteristics of P. talassica × P. euphratica Roots, Stems and Leaves

#### 4.3.1. Anatomical Features of *P. talassica* × *P. euphratica* Leaves

Salinity affected anatomical and morphological characters along with physiological parameters [85]. Plants generally develop salt-resistance mechanisms and unique structures to survive under high saline stress conditions [90,91]. The studying of changes in leaf anatomy is an appropriate way to research into abiotic stress situations, including salt stress [18]. In this study, compared with the CK, NaCl-stress treatments significantly increased the palisade tissue thickness and leaf thickness, but significantly decreased spongy tissue thickness, of *P. talassica* × *P. euphratica* leaves. Increases in leaf thickness can be induced by the exposure of roots to high concentrations of NaCl [80,86]. Increasing the palisade tissue thickness and leaf thickness of *P. talassica* × *P. euphratica* may be structural changes that help to resist salt stress. Similar to our findings, Mohsen et al., on guava, Salem et al., on apple seedling, and Sherin and Pfeiffer et al., on olive, found that increasing salinity levels were accompanied by increases in the thicknesses of leaf blades [92,93,94,95].

Interestingly, contrary to our findings, E Karimi et al. (2012) observed an increase in spongy mesophyll thickness due to salt stress, along with a slight reduction in palisade mesophyll thickness [96]. Therefore, the response of *P. talassica* × *P. euphratica* to external adversity stress can be significantly reflected in the structures of leaves first, which then affects their physiological and biochemical responses.

#### 4.3.2. Anatomical Features of *P. talassica* × *P. euphratica* Stems

In the stem, adaptive features include a thickening epidermis and increasing phloem area [97]. In our current research, compared with the CK, the anatomical characteristics of *P. talassica* × *P. euphratica* seedling stems changed significantly under different concentrations of NaCl. The NaCl-stress increased stem epidermal thickness, cortex thickness, phloem and xylem thickness and pith diameter. Both the cortical and pith increased in the salt range population. These tissues can enhance the storage capacity, which is crucial under unfavorable moisture conditions [98]. Similar to our findings, the stem cortex thickness of the Palestinian tomato (*Solanum lycopersicon*) increases significantly at 100 mmol/L NaCl compared with the control [84]. Moreover, Younis et al. observed that the phloem cell areas and pith cell areas of stems increase under salt stress [99], which was similar to our findings. This increased succulence in stems may aid to store additional water, and hence, increase survival, under harsh environmental conditions [100]. Furthermore, epidermal thickness greatly increased, indicating increased adaptability because a thick epidermis is a characteristic feature of salt-tolerant species. This characteristic is critical when moisture availability is limited because a thick epidermis is capable of checking water loss through stems [101,102,103,104].

#### 4.3.3. Anatomical Features of *P. talassica* × *P. euphratica* Roots

The effects of salinity on the root anatomy of plants have been reported previously [105,106,107]. In this study, root xylem thickness, phloem thickness, xylem vessel number, xylem vessel diameter and root cross-sectional diameter all showed a significant upward trend as the NaCl concentration increased. In addition, the variation in the diameter of the xylem vessels influenced the salinity tolerance. The diameter of the xylem is considerably larger in salt-treated plants [108]. Contrary to our findings, Maryani et al. showed that increases in NaCl concentrations caused decreases in root diameters and xylem tissue thickness. However, in their findings, similar to our results, high concentrations of NaCl tended to increase the diameters of xylem vessels in roots [109]. The diameters of the xylem vessels affect the water transport capacity [110]: the larger the diameters of the xylem vessels, the stronger the water transport capacity.

## 5. Conclusions

The growth, physiological and photosynthetic characteristics of *P. talassica* × *P. euphratica* seedlings, as well as root, stem and leaf anatomy are highly dependent on soil moisture and salinity conditions. Under low-salt stress (200 mmol/L NaCl), these indicators were limited, but *P. talassica* × *P. euphratica* could adapt to the low-salt stress by changeing the leaf area, leaf thickness, root length, root branch number, plant height, ground diameter, crown width and biomass allocation, improving water use efficiency, changing stomatal density and size, increasing the content of Pro, MDA, soluble sugar and soluble protein, increasing the activities of three antioxidant enzymes (SOD, POD and CAT), and changing the anatomical structure of *P. talassica* × *P. euphratica*, etc. A series of strategies are used to adapt to the salt environment, so that they can grow normally, and to a certain extent promote their growth. *P. talassica* × *P. euphratica* can survive and reproduce at a concentration of more than 200 mmol/L NaCl, which proves that it has salt [11,69,111]. However, under high salt stress (400 mmol/L NaCl), various growth parameters of *P. talassica* × *P. euphratica* seedlings decreased, biomass accumulation decreased, photosynthesis decreased significantly, WUE decreased and antioxidant enzyme activities decreased, indicating that the above mechanisms cannot offset the salt coercion effect.

In accordance with the related growth, physiological and photosynthetic characteristics of *P. talassica* × *P. euphratica* in response to salt stress, as well as the anatomical structural characteristics of roots, stems and leaves, *P. talassica* × *P. euphratica* appeared to have a certain salt tolerance, and moderate salt stress was beneficial to its growth. The growth and physiological characteristics of *P. talassica* × *P. euphratica* in response to salt stress, as well as changes in its structures, were adaptive mechanisms to cope with salt stress. This study provides reliable physiological, biochemical and structural indicators for exploring the salt tolerance mechanism of *P. talassica* × *P. euphratica*, and provides a certain reference for the improvement and sustainable development of soil saline-alkali land in Xinjiang in the future.

## Figures and Tables

**Figure 1 plants-11-03025-f001:**
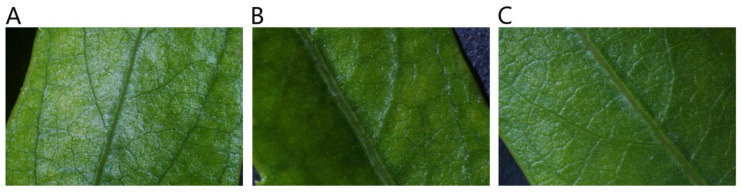
Effects of different concentrations of NaCl-stress treatments on salt secretion of *P. talassica* × *P. euphratica* leaf surfaces. (**A**) CK (1×); (**B**) 200 mmol/L NaCl (1×); (**C**) 400 mmol/L NaCl (1×).

**Figure 2 plants-11-03025-f002:**
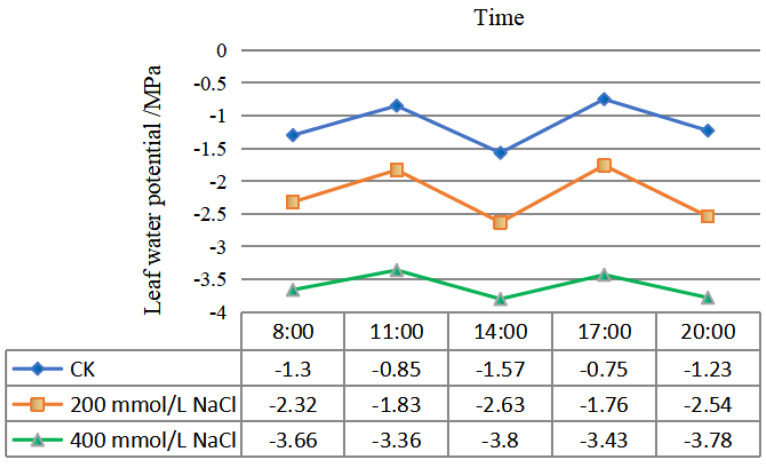
Diurnal variation curve of the *P. talassica* × *P. euphratica* leaf water potential.

**Figure 3 plants-11-03025-f003:**
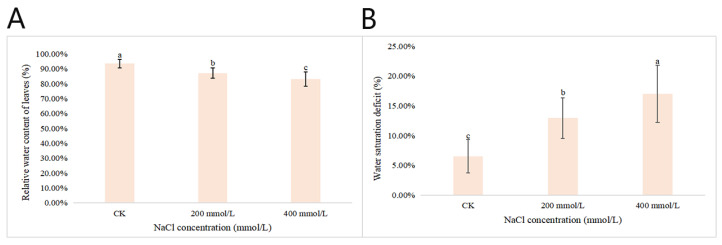
Effects of different concentrations of NaCl-stress treatments on the relative water contents and water saturation deficits of *P. talassica* × *P. euphratica* leaves. (**A**) Effects of different concentrations of NaCl-stress treatments on the relative water contents in leaves of *P. talassica* × *P. euphratica*; (**B**) Effects of different concentrations of NaCl-stress treatments on the water saturation deficits in leaves of *P. talassica* × *P. euphratica*. Bars represent means and standard deviations of biological replicates. Letters above bars indicate statistical signifcance by one-way ANOVA, different lowercase letters indicate significant differences, *p* < 0.05.

**Figure 4 plants-11-03025-f004:**
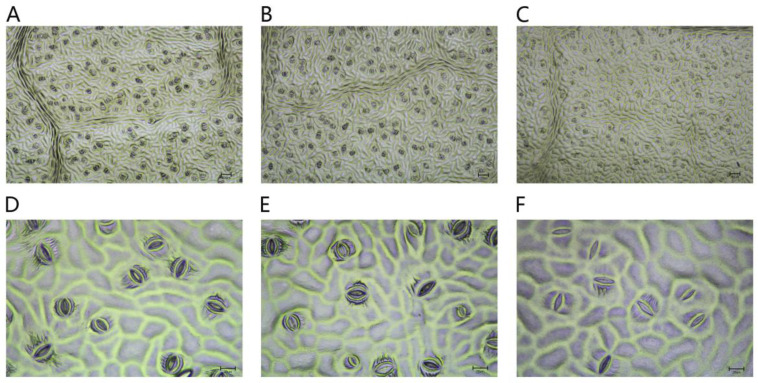
Effects of different concentrations of NaCl-stress treatments on the size, and the opening and closing, of *P. talassica* × *P. euphratica* leaf stomata. (**A**–**E**) Microstructures of leaf stomata in leaves: (**A**) in their natural state (×10); (**B**) treated with 200 mmol/L NaCl (×10); (**C**) treated with 400 mmol/L NaCl (×10); (**D**) in their natural state (×40); (**E**) treated with 200 mmol/L NaCl (×40); (**F**) treated with 400 mmol/L NaCl (×40).

**Figure 5 plants-11-03025-f005:**
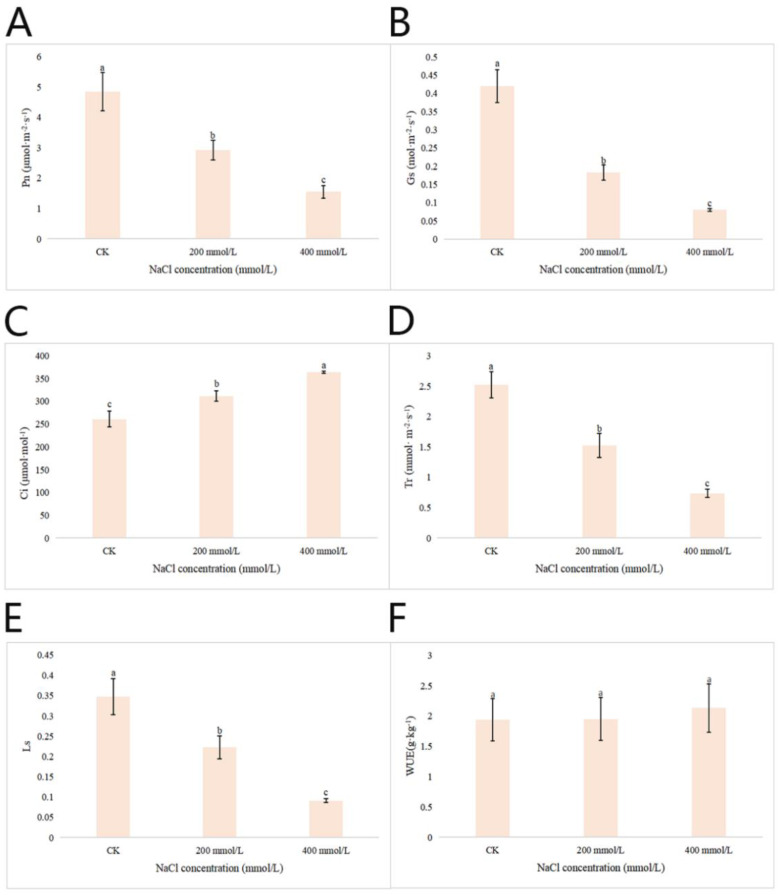
Effects of different concentrations of NaCl-stress treatments on the photosynthetic physiology of *P. talassica* × *P. euphratica*. (**A**) Effects of different concentrations of NaCl-stress treatments on the net photosynthetic rate in leaves of *P. talassica* × *P. euphratica*; (**B**) Effects of different concentrations of NaCl-stress treatments on the stomatal conductance in leaves of *P. talassica* × *P. euphratica*; (**C**) Effects of different concentrations of NaCl-stress treatments on the intercellular CO_2_ concentration in leaves of *P. talassica* × *P. euphratica*; (**D**) Effects of different concentrations of NaCl-stress treatments on the transpiration rate in leaves of *P. talassica* × *P. euphratica*; (**E**) Effects of different concentrations of NaCl-stress treatments on the stomatal limitation in leaves of *P. talassica* × *P. euphratica*; (**F**) Effects of different concentrations of NaCl-stress treatments on the the water use efficiency in leaves of *P. talassica* × *P. euphratica*. Bars represent means and standard deviations of biological replicates. Letters above bars indicate statistical signifcance by one-way ANOVA, different lowercase letters indicate significant differences, *p* < 0.05.

**Figure 6 plants-11-03025-f006:**
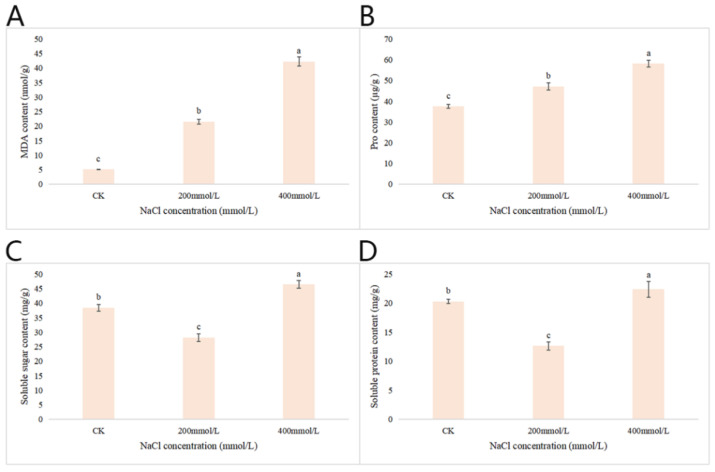
Effects of different concentrations of NaCl-stress treatments on the contents of MDA, Pro, soluble sugar and soluble protein in *P. talassica* × *P. euphratica* leaves. (**A**) Effects of different concentrations of NaCl-stress treatments on the malondialdehyde content in leaves of *P. talassica* × *P. euphratica*; (**B**) Effects of different concentrations of NaCl-stress treatments on the proline content in leaves of *P. talassica* × *P. euphratica*; (**C**) Effects of different concentrations of NaCl-stress treatments on the soluble sugar content in leaves of *P. talassica* × *P. euphratica*; (**D**) Effects of different concentrations of NaCl-stress treatments on the soluble protein content in leaves of *P. talassica* × *P. euphratica*. Bars represent means and standard deviations of biological replicates. Letters above bars indicate statistical signifcance by one-way ANOVA, different lowercase letters indicate significant differences, *p* < 0.05.

**Figure 7 plants-11-03025-f007:**
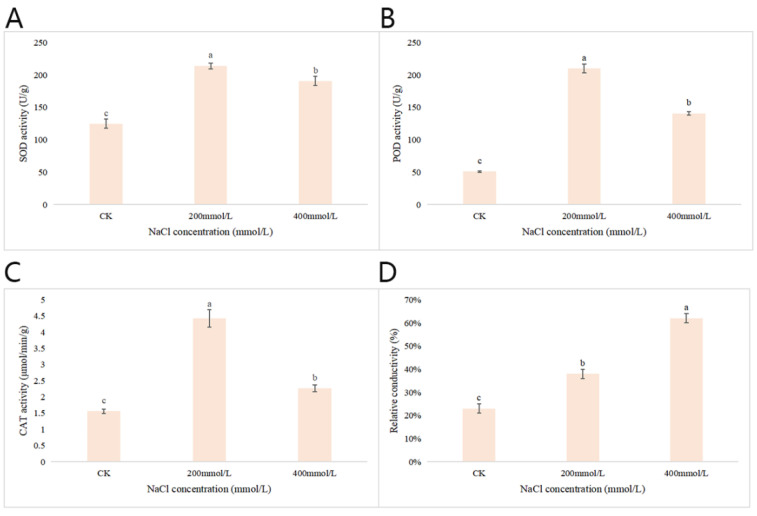
Effects of different concentrations of NaCl-stress treatments on antioxidant enzyme (SOD, POD and CAT) activities and the relative electrical conductivity in leaves. (**A**) Effects of different concentrations of NaCl-stress treatments on the activity of superoxide dismutase in *P. talassica* × *P. euphratica* leaves; (**B**) Effects of different concentrations of NaCl-stress treatments on the activity of peroxidase in *P. talassica* × *P. euphratica* leaves; (**C**) Effects of different concentrations of NaCl-stress treatments on the activity of catalase in *P. talassica* × *P. euphratica* leaves; (**D**) Effects of different concentrations of NaCl-stress treatments on the relative conductivity in *P. talassica* × *P. euphratica* leaves. Bars represent means and standard deviations of biological replicates. Letters above bars indicate statistical signifcance by one-way ANOVA, different lowercase letters indicate significant differences, *p* < 0.05.

**Figure 8 plants-11-03025-f008:**
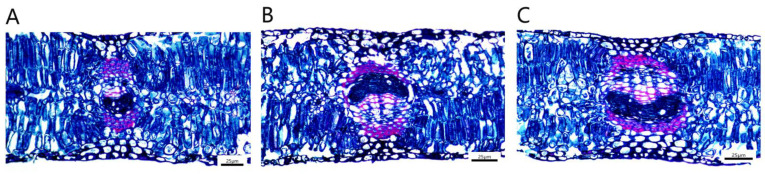
Microstructures of leaves treated with different concentrations of NaCl-stress treatments (20×). (**A**) CK (20×); (**B**) 200 mmol/L NaCl (20×); (**C**) 400 mmol/L NaCl (20×).

**Figure 9 plants-11-03025-f009:**
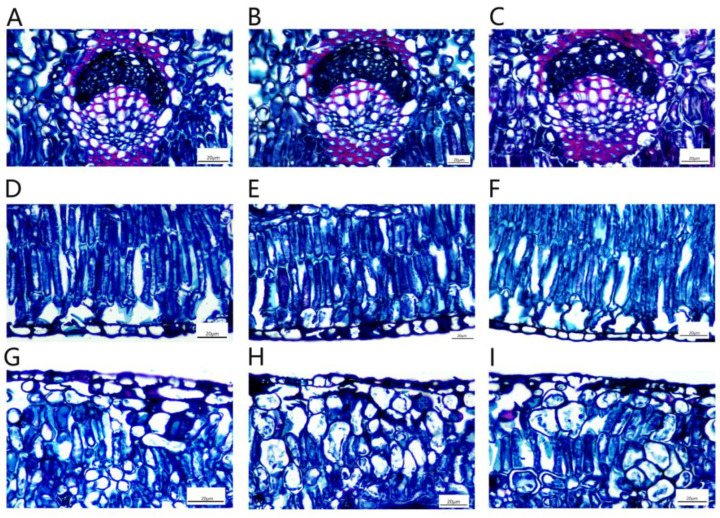
Local microstructure of leaves treated with different concentrations of NaCl-stress treatments (40×). (**A**–**C**) Microstructures of leaf vascular bundles: (**A**) under natural condition (40×); (**B**) in leaves treated with 200 mmol/L NaCl (40×); (**C**) in leaves treated with 400 mmol/L NaCl (40×). (**D**–**F**) Microstructures of leaf palisade tissues: (**D**) under natural condition (40×); (**E**) in leaves treated with 200 mmol/L NaCl (40×); (**F**) in leaves treated with 400 mmol/L NaCl (40×). (**G**–**I**) Microstructures of leaf sponges: (**G**) under natural conditions (40×); (**H**) treated with 200 mmol/L NaCl (40×); (**I**) treated with 400 mmol/L NaCl (40×).

**Figure 10 plants-11-03025-f010:**
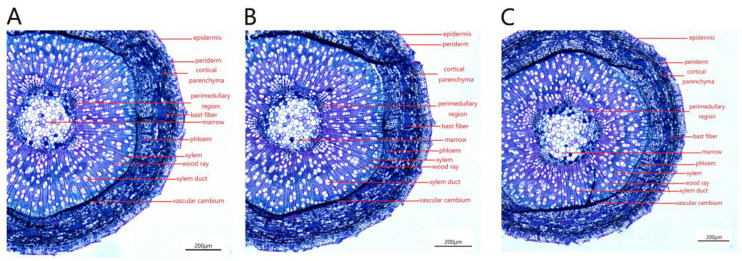
Microstructures of stems treated with different concentrations of NaCl-stress treatments. (**A**) CK (4×); (**B**) 200 mmol/L NaCl (4×); (**C**) 400 mmol/L NaCl (4×).

**Figure 11 plants-11-03025-f011:**
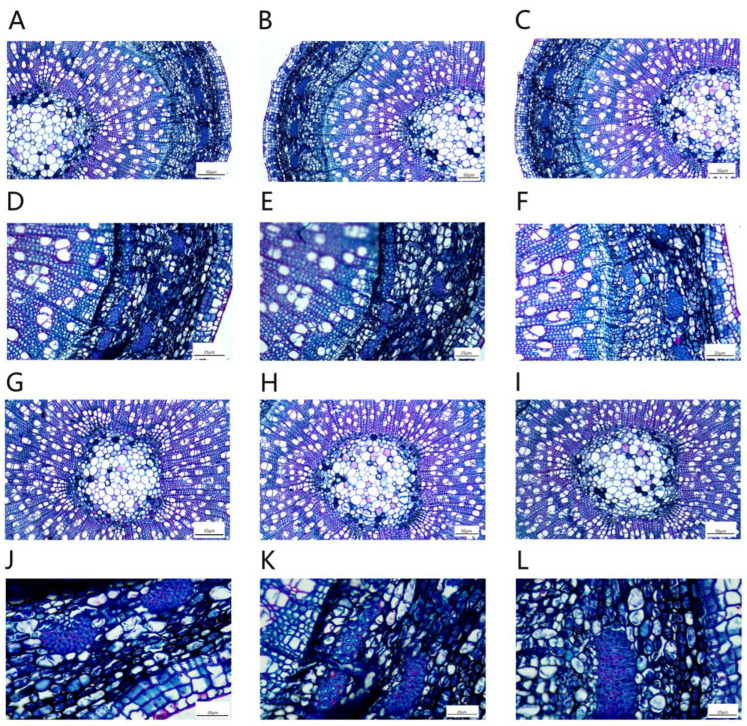
Local microstructures of *P. talassica* × *P. euphratica* stems treated with different concentrations of NaCl-stress treatments. (**A**) Microscopic local structure of natural stems (×10); (**B**) Microscopic local structure of the stems under 200 mmol/L NaCl-stress treatment (×10); (**C**) Microscopic local structure of the stem under 400 mmol/L NaCl-stress treatment (×10); (**D**) Microstructure of the phloem of the stems in the natural state (×20); (**E**) Microstructure of the phloem of the stems under 200 mmol/L NaCl-stress treatment (×20); (**F**) Microstructure of the phloem of the stems under 400 mmol/L NaCl-stress treatment (×20); (**G**) Natural stem pith microstructure (×10); (**H**) Stem pith microstructure under 200 mmol/L NaCl-stress treatment (×10); (**I**) Stem pith microstructure under 400 mmol/L NaCl-stress treatment (×10); (**J**) Microstructure of natural stems phloem fibers (×40); (**K**) Microstructure of stems phloem fibres under 200 mmol/L NaCl-stress treatment (×40); (**L**) Microstructure of stem phloem fibres under 400 mmol/L NaCl-stress treatment (×40).

**Figure 12 plants-11-03025-f012:**
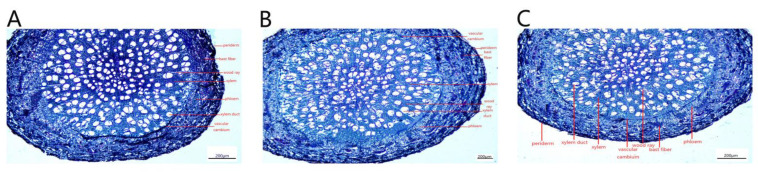
Microstructural characteristics of *P. talassica* × *P. euphratica* secondary root structures under different concentrations of NaCl-stress treatments. (**A**) CK (4×); (**B**) 200 mmol/L NaCl (4×); (**C**) 400 mmol/L NaCl (4×).

**Figure 13 plants-11-03025-f013:**
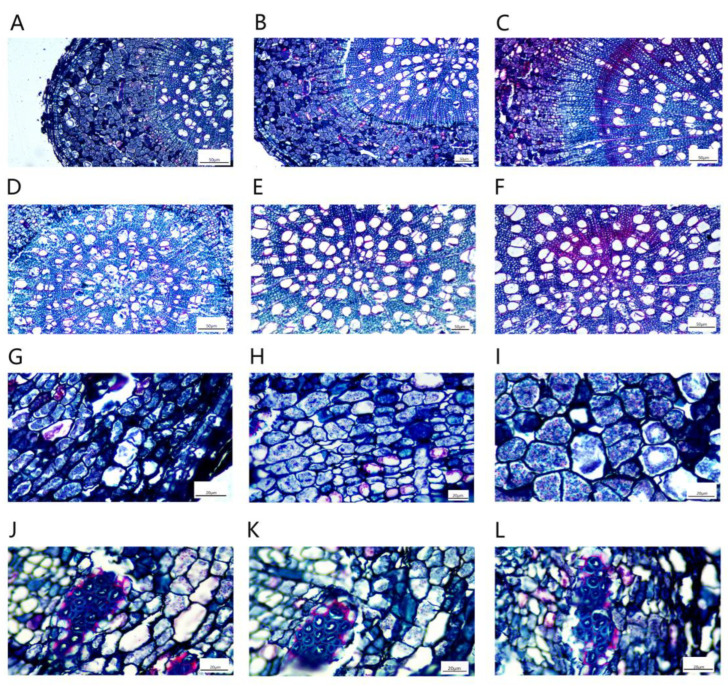
Microstructural characteristics of *P. talassica* × *P. euphratica* secondary root structures under different concentrations of NaCl-stress treatments. (**A**) Local microstructure of natural roots (×10); (**B**) Local root microstructure under 200 mmol/L NaCl-stress treatment (×10); (**C**) Local root microstructure under 400 mmol/L NaCl-stress treatment (×10); (**D**) Microstructure of natural root xylem (×10); (**E**) Root xylem microstructure under 200 mmol/L NaCl-stress treatment (×10); (**F**) Root xylem microstructure under 400 mmol/L NaCl-stress treatment (×10); (**G**) Natural root phloem microstructure (×40); (**H**) Microscopic structure of root phloem under 200 mmol/L NaCl-stress treatment (×40); (**I**) Microscopic structure of root phloem under 400 mmol/L NaCl-stress treatment (×40); (**J**) Natural root phloem fibers microstructure (×40); (**K**) Root phloem fibers microstructure under 200 mmol/L NaCl-stress treatment (×40); (**L**) Root phloem fibers microstructure under 400 mmol/L NaCl-stress treatment (×40).

**Table 1 plants-11-03025-t001:** Effects of different concentrations of NaCl-stress treatments on leaf length, leaf width, leaf area, leaf number and the survival rate of *P. talassica* × *P. euphratica*.

NaCl	Leaf Area	Leaf Length	Average Leaf Width	Number of Leaves	Survival Rate
(mmol/L)	(cm^2^)	(cm)	(cm)	(Pieces)	(%)
CK	10.07 ± 0.18 b	10.24 ± 0.22 b	0.90 ± 0.07 a	17.60 ± 1.14 b	100.00 ± 0.00 a
200	10.60 ± 0.35 a	11.04 ± 0.24 a	0.98 ± 0.16 a	19.80 ± 0.84 a	100.00 ± 0.00 a
400	8.56 ± 0.23 c	7.10 ± 0.16 c	0.67 ± 0.04 b	15.20 ± 0.84 c	91.60 ± 1.14 b

Note: The data in the table represent means ± standard errors, and different lowercase letters indicate significant differences (*p* < 0.05).

**Table 2 plants-11-03025-t002:** Effects of different concentrations of NaCl-stress treatments on *P. talassica* × *P. euphratica* biomass.

NaCl	Aboveground Biomass	Underground Biomass	Whole Plant Biomass	Root to Shoot Ratio
(mmol/L)	(g)	(g)	(g)	
CK	7.49 ± 0.53 a	4.33 ± 0.30 b	11.82 ± 0.70 a,b	0.58 ± 0.04 b
200	7.91 ± 0.30 a	5.41 ± 0.51 a	13.32 ± 0.58 a	0.68 ± 0.07 a
400	6.12 ± 0.47 b	4.49 ± 0.46 b	10.61 ± 0.93 b	0.73 ± 0.02 a

Note: The data in the table represent means ± standard errors, and different lowercase letters indicate significant differences (*p* < 0.05).

**Table 3 plants-11-03025-t003:** Effects of different concentrations of NaCl-stress treatments on *P. talassica* × *P. euphratica* root vitality.

NaCl	Length	Surface Area	Volume	Average Diameter	Number of Forks
(mmol/L)	(cm)	(cm^2^)	(cm^3^)	(mm)	
CK	1080.35 ± 81.94 b	459.76 ± 58.56 b	39.54 ± 2.79 b	0.99 ± 0.03 b,c	3475.80 ± 328.77 b
200	1703.38 ± 144.41 a	620.01 ± 57.51 a	51.06 ± 4.17 a	1.22 ± 0.31 a,b	6068.40 ± 652.26 a
400	769.91 ± 171.04 c	330.47 ± 51.10 c	29.82 ± 1.37 c	0.87 ± 0.08 b,c	2574.40 ± 572.54 c

Note: The data in the table represent means ± standard errors, and different lowercase letters indicate significant differences (*p* < 0.05).

**Table 4 plants-11-03025-t004:** Effects of different concentrations of NaCl-stress treatments on the plant height, ground diameter and crown width of *P. talassica* × *P. euphratica*.

Growth Indexes (cm)	NaCl (mmol/L)		
	CK	200 mmol/L	400 mmol/L
Crown width before NaCl treatment (cm)	12.88 ± 0.18 a	12.84 ± 0.24 a	12.87 ± 0.19 a
Crown width 45 d after NaCl treatment (cm)	15.77 ± 0.25 b	16.97 ± 0.29 a	14.92 ± 0.30 c
Relative crown growth (cm)	2.89 ± 0.17 b	4.13 ± 0.25 a	2.05 ± 0.16 c
Plant height before NaCl treatment (cm)	9.97 ± 0.32 a	10.02 ± 0.32 a	10.04 ± 0.27 a
Plant height 45 d after NaCl treatment (cm)	10.68 ± 0.32 b	11.06 ± 0.32 a	10.45 ± 0.23 b
Relative plant height growth (cm)	0.71 ± 0.07 b	1.04 ± 0.07 a	0.41 ± 0.10 c
Ground diameter before NaCl treatment (cm)	0.71 ± 0.02 a	0.71 ± 0.02 a	0.70 ± 0.02 a
Ground diameter 45 d after NaCl treatment (cm)	1.20 ± 0.03 b	1.35 ± 0.07 a	1.08 ± 0.04 c
Relative ground diameter growth (cm)	0.49 ± 0.01 b	0.65 ± 0.05 a	0.39 ± 0.02 c

Note: The data in the table represent means ± standard errors, and different lowercase letters indicate significant differences (*p* < 0.05).

**Table 5 plants-11-03025-t005:** Effects of different concentrations of NaCl-stress treatments on the *P. talassica* × *P. euphratica* leaf chlorophyll content.

NaCl	Chlorophyll a Content	Chlorophyll b Content	Chlorophyll a/b	Total Chlorophyll Content	Carotene Content
(mmol/L)	(mg/g)	(mg/g)		(mg/g)	(mg/g)
CK	13.80 ± 1.37 a	4.40 ± 0.47 a	3.14 ± 0.07 a	18.20 ± 1.84 a	3.26 ± 0.30 a
200	9.19 ± 0.82 b	2.93 ± 0.31 b	3.14 ± 0.14 a	12.13 ± 1.10 b	2.13 ± 0.16 b
400	4.48 ± 0.72 c	1.49 ± 0.25 c	3.01 ± 0.15 b	5.98 ± 0.97 c	1.18 ± 0.11 c

Note: The data in the table represent means ± standard errors, and different lowercase letters indicate significant differences (*p* < 0.05).

**Table 6 plants-11-03025-t006:** Effects of different concentrations of NaCl-stress treatments on the size, and the opening and closing, of stomata in *P. talassica* × *P. euphratica* leaves.

NaCl	Stomatal Length	Stomatal Width	Stomatal Area	Stomatal Density	Stomatal Area Index	Proportion of Opened Stomata
(mmol/L)	(μm)	(μm)	(μm^2^)	(number mm^−2^)	(%)	(%)
CK	32.98 ± 0.69 a	14.92 ± 1.15 a	328.23 ± 5.41 a	573.12 ± 14.52 a	0.18 ± 0.01 a	0.44 ± 0.02 a
200	31.20 ± 0.82 a	12.08 ± 0.49 b	283.87 ± 6.63 b	474.31 ± 23.88 b	0.14 ± 0.01 b	0.32 ± 0.02 b
400	22.65 ± 2.10 b	8.22 ± 0.98 c	123.34 ± 8.68 c	350.99 ± 12.31 c	0.04 ± 0.01 c	0.15 ± 0.03 c

Note: The data in the table represent means ± standard errors, and different lowercase letters indicate significant differences (*p* < 0.05).

**Table 7 plants-11-03025-t007:** Effects of different concentrations of NaCl-stress treatments on the anatomical structures of *P. talassica* × *P. euphratica* leaves.

NaCl	Palisade Tissue Thickness	Spongy Tissue Thickness	Blade Thickness	CP	The Tightness of the Blade Structure
(mmol/L)	(μm)	(μm)	(μm)		
CK	274.12 ± 12.42 c	423.31 ± 12.06 a	762.85 ± 17.90 a	0.66 ± 0.04 c	0.36 ± 0.01 c
200	365.40 ± 17.77 b	377.43 ± 13.68 b	790.79 ± 16.36 b	0.83 ± 0.03 b	0.46 ± 0.02 b
400	451.25 ± 10.45 a	319.91 ± 14.45 c	812.78 ± 20.25 c	1.30 ± 0.04 a	0.56 ± 0.01 a

Note: The data in the table represent means ± standard errors, and different lowercase letters indicate significant differences (*p* < 0.05).

**Table 8 plants-11-03025-t008:** Effects of different concentrations of NaCl-stress treatments on the anatomical structures of *P. talassica* × *P. euphratica* stems.

NaCl	Epidermal Thickness	Cortical Thickness	Phloem Thickness	Xylem Thickness	Pith Diameter
(mmol/L)	(μm)	(μm)	(μm)	(μm)	(μm)
CK	20.58 ± 1.75 c	59.90 ± 4.05 c	325.85 ± 8.14 c	761.71 ± 24.96 c	921.67 ± 13.22 c
200	26.13 ± 1.17 b	74.94 ± 3.82 b	402.36 ± 9.81 b	819.33 ± 6.30 b	1026.27 ± 28.47 b
400	29.66 ± 0.84 a	92.32 ± 1.98 a	479.27 ± 22.23 a	859.22 ± 17.34 a	1169.68 ± 48.38 a

Note: The data in the table represent means ± standard errors, and different lowercase letters indicate significant differences (*p* < 0.05).

**Table 9 plants-11-03025-t009:** Effects of different concentrations of NaCl-stress treatments on the root secondary structures of *P. talassica* × *P. euphratica*.

NaCl	Peripheral Thickness	Phloem Thickness	Xylem Thickness	Xylem Vessel Diameter	Root Cross Section Diameter
(mmol/L)	(μm)	(μm)	(μm)	(μm)	(μm)
CK	19.95 ± 1.32 a,b	629.44 ± 52.66 c	1739.67 ± 88.77 c	121.26 ± 8.09 c	2510.32 ± 106.45 c
200	19.42 ± 1.14 b	786.87 ± 37.04 b	2140.24 ± 112.59 b	167.14 ± 6.28 b	3113.67 ± 132.22 b
400	21.19 ± 1.75 a	960.38 ± 49.10 a	3405.23 ± 214.93 a	234.69 ± 12.66 a	4621.50 ± 231.73 a

Note: The data in the table represent means ± standard errors, and different lowercase letters indicate significant differences (*p* < 0.05).

## Data Availability

Not applicable.

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
