# Peer review of "Effects of NaCl Stress on the Growth, Physiological Characteristics and Anatomical Structures of Populus talassica × Populus euphratica Seedlings"

_plants, 2022, doi:10.3390/plants11223025_

Round 1
Reviewer 1 Report
The manuscript covers the problem of plant adaptation to salinity.
The way the topic and data are presented should again be substantially rethought and improved.
I propose to:
1. analyse and revise the literature presented (mainly in the 'introduction' and 'discussion' sections) so that it presents and justifies the issues related to salinity and the plant under study, i.e. Populus talassica × Populus euphratica. The text is dominated by literature related to halophytes (why?).
There is no enough information of how glycophytes adapt in the context of salinity.
2. both the 'introduction' and ' discussion' sections do not reflect/are not representative of the results presented
3. discuss/describe the experimental facility studied, indicate why this facility was chosen for analysis under salinity stress conditions
4. shorten the text considerably, in particular re-examine the data presented, the way they are presented and draw attention to the relationships between the parameters studied
- maybe combine tables, figures into larger segments?
- the authors describe the results as the dynamics of change - dynamics was not measured
- clarify frequently used indefinite terms, e.g. '(...) first increased and then decreased (...)'
- why plants were watered with deionized water?
5. correct descriptions of figures since they are not visible
6. Complete the statistics in Fig. 2
7. replace the images shown (Figs 1, 4, 8, 9, 10, 11) with images taken at higher magnification and resolution
8. the discussion is an abbreviated repetition of the description of the results. The discussion should be reconsidered. The results should be discussed with the current literature - including that related to Populus species.
I suggest highlighting the relationships between parameters and emphasising the new developments of this work.
I suggest:
- use the website:
www.sussex.ac.uk/affiliates/halophytes/
- read articles such as:
· Liu, Y.; Han, Z.J.; Su, M.X.; Zhang, M. Transcriptomic Profile Analysis of Populus talassica Populus euphratica Response and Tolerance under Salt Stress Conditions. Genes 2022, 13, 1032. https://doi.org/10.3390/genes13061032
· S. Chen,A. Polle Salinity tolerance of Populus. Plant Biology 12 (2010) 317–333 https://doi.org/10.1111/j.1438-8677.2009.00301.x
Zhang C, Luo W, Li Y, Zhang X, Bai X, Niu Z, Zhang X, Li Z and Wan D (2019) Transcriptomic Analysis of Seed Germination Under Salt Stress in Two Desert Sister Species (Populus euphratica and P. pruinosa). Front. Genet. 10:231. doi: 10.3389/fgene.2019.00231
· Surówka E, et al. . Tocopherols mutual balance is a key player for maintaining Arabidopsis thaliana growth under salt stress. Plant Physiol Biochem. 2020 Nov;156:369-383. doi: 10.1016/j.plaphy.2020.09.008.
· Liu et al. BMC Plant Biology (2019) 19:367; https://doi.org/10.1186/s12870-019-1952-2
· Surówka E. et al. ROS-Scavengers, Osmoprotectants and Violaxanthin De-Epoxidation in Salt-Stressed Arabidopsis thaliana with Different Tocopherol Composition. Int. J. Mol. Sci. 2021, 22, 11370. https://doi.org/10.3390/ijms222111370
Author Response
Dear Editor and Reviewers:
Thank you very much for your comments and suggestions.
Those comments are all valuable and very helpful for revising and improving our paper, as well as the important guiding significance to our researches. We have studied comments carefully and have made correction which we hope meet with approval. Revised portion are markedin red in the paper. The main corrections in the paper and the responds to the reviewer’s comments are as flowing:
Responds to the reviewer’s comments:
Reviewer 1:
1. Comment: analyse and revise the literature presented (mainly in the 'introduction' and 'discussion' sections) so that it presents and justifies the issues related to salinity and the plant under study, i.e. Populus talassica × Populus euphratica. The text is dominated by literature related to halophytes (why?).
There is no enough information of how glycophytes adapt in the context of salinity.
Response: Thank you very much for your valuable comments. I have revised the "Introduction" and "Discussion" sections to compare Populus talassica × Populus euphratica and other plants of the genus Populus L., and the references have also added many other literatures related to other plants of the genus Populus L.. I also added in the discussion section how glycophytes adapt in the context of salinity .
2. Comment:both the 'introduction' and 'discussion' sections do not reflect/are not representative of the results presented
Response: Thank you very much for your valuable comments, I have revised the 'introduction' and 'discussion' sections.
3.Comment: discuss/describe the experimental facility studied, indicate why this facility was chosen for analysis under salinity stress conditions
Response: Thank you very much for your valuable comments. Most studies on the salt tolerance of poplar species have used young rooted cuttings for experimental material. The choice of younger plants is required by the fast growth rate of most poplars: trees more than 1 or 2 years old become too large for easy handling in the laboratory or greenhouse . The annual potted Populus talassica × Populus euphratica seedlings were chosen as the experimental material to be treated under the condition of salt stress because the potted seedlings are easy to control variables and further process, observe and measure.
4.Comment:shorten the text considerably, in particular re-examine the data presented, the way they are presented and draw attention to the relationships between the parameters studied
- maybe combine tables, figures into larger segments?
- the authors describe the results as the dynamics of change - dynamics was not measured
- clarify frequently used indefinite terms, e.g. '(...) first increased and then decreased (...)'
- why plants were watered with deionized water?
Response: Thank you very much for your valuable comments, I've re-examined my data and how they're presented, and I've removed the ambiguous term "(...) increases first, then decreases(...)" from the results.The reason for choosing deionized water to water the plants is to prevent the ions in the tap water from affecting the results of the experimental treatments and to ensure the accuracy of the experiments.
5.Comment:correct descriptions of figures since they are not visible
Response: Thank you very much for your valuable comments, I have re-described all the pictures correctly.
6.Comment:Complete the statistics in Fig. 2
Response: Thank you very much for your valuable comments, I have completed the statistics in Figure 2.
7.Comment:replace the images shown (Figs 1, 4, 8, 9, 10, 11) with images taken at higher magnification and resolution
Response: Thank you very much for your valuable comments, I have replaced the images shown (Figures 1, 4, 8, 9, 10, 11) with higher magnification and resolution images.
8.Comment:the discussion is an abbreviated repetition of the description of the results. The discussion should be reconsidered. The results should be discussed with the current literature - including that related to Populusspecies.
I suggest highlighting the relationships between parameters and emphasising the new developments of this work.
Response: Thank you very much for your valuable comments, I have reorganized the Discussion section and discussed the findings with the existing literature on poplars. I have also highlighted the relationship between the parameters and highlighted new developments in this work.
Best wishes,
Ying Liu

Reviewer 2 Report
There are some issues in the paper, so I have some suggestions:
In my opinion, I think the title of the paper should be: ''Effects of NaCl stress on the growth, physiological characteristics and anatomical structures of Populus talassica × Populus euphratica seedlings''.
Instead of using the term ''cultivar'' (lines 58 and 59), I suggest "seedling population'' since the seed production is not controlled and the respective seedling population is not yet officially recognized as a cultivated variety or a cultivar.
There are some recent papers, on almost the same subject, with some common authors, which are not cited i.e.:
SUN Yang, HAN Zhanjiang, SHI Jianyin, TANG Hongyou, LI Kang. Effects of Salt Environment on Growth and Photosynthetic Characteristics of Populus talassica × P. euphratica[J]. Xinjiang Agricultural Sciences, 2021, 58(4): 634-642.
Tang Hongyou, Shi Jianyin, Han Zhanjiang, Bai Menghan, Sun Yang, Zhang Min. Physiological Response of Populus talassica× P. euphratica to Na2SO4 Environment[J]. Chinese Agricultural Science Bulletin, 2021, 37(4): 38-42.
I suggest mentioning them and comparing the results obtained.
Physiological ResponseSO4Environment
Author Response
Dear Editor and Reviewers:
Thank you very much for your comments and suggestions.
Those comments are all valuable and very helpful for revising and improving our paper, as well as the important guiding significance to our researches. We have studied comments carefully and have made correction which we hope meet with approval. Revised portion are markedin red in the paper. The main corrections in the paper and the responds to the reviewer’s comments are as flowing:
Responds to the reviewer’s comments:
Reviewer 2:
1. Comment: In my opinion, I think the title of the paper should be: ''Effects of NaCl stress on the growth, physiological characteristics and anatomical structures of Populus talassica × Populus euphratica seedlings''.
Response: Thank you very much for your valuable comments, I have changed the title of the paper to: ''Effects of NaCl stress on the growth, physiological characteristics and anatomical structures of Populus talassica × Populus euphratica seedlings''.
2.Comment:Instead of using the term ''cultivar'' (lines 58 and 59), I suggest "seedling population'' since the seed production is not controlled and the respective seedling population is not yet officially recognized as a cultivated variety or a cultivar.
Response: Thank you very much for your valuable comments, I have made changes.
3.Comment:There are some recent papers, on almost the same subject, with some common authors, which are not cited i.e.:
SUN Yang, HAN Zhanjiang, SHI Jianyin, TANG Hongyou, LI Kang. Effects of Salt Environment on Growth and Photosynthetic Characteristics of Populus talassica × P. euphratica[J]. Xinjiang Agricultural Sciences, 2021, 58(4): 634-642.
Tang Hongyou, Shi Jianyin, Han Zhanjiang, Bai Menghan, Sun Yang, Zhang Min. Physiological Response of Populus talassica× P. euphratica to Na2SO4 Environment[J]. Chinese Agricultural Science Bulletin, 2021, 37(4): 38-42.
I suggest mentioning them and comparing the results obtained.
Response: Thank you very much for your valuable comments, I have included these references.
Best wishes,
Ying Liu

Reviewer 3 Report
Dear authors,
The topic of the paper fits well with the scope of the journal. The present paper is well structured, and even non-expert readers can easily follow the content, however, in my opinion, scientific significance and novelty are relatively low.
Despite the comprehensibility and clarity, certain errors give the impression before the article is published. These errors need to be addressed before the article is published.
ABSTRACT:
L13: I don't think the word "gradient" is appropriate here. I will use word "treatment".
MATERIAL AND METHODS:
In some places, the abbreviations which are used letter in the Results and Discussion sections in M&M are missing or there are abbreviations without explanation. Please revise through the M&M section.
L194-195: Why uppercase letters are used? The same for L198.
RESULTS:
For all tables in Results section please, arrange the text alignment.
Table 3: Please check the units.
Subtitle 3.2.1 is the same as 3.1.1.
There is no explanation which treatment represents foto A, B, and C in Figure 1.
L285: remove "and"
Figure 3B: Water saturation deficit is not mentioned in M&M section.
L342-346: The same statements are written twice. Please delete the first two sentences or delete the third one.
L360-364: Both sentences say very similar things. Please reformat the text to avoid such duplications. Do this throughout the Results chapter. In this way, you can greatly shorten a text that is relatively long.
Figure 8: explanation of which treatment represents foto A, B, and C is missing. The labels explained in the Notes below the figure are not visible in the pictures (very poor resolution).
Figure 10: very poor resolution for text in the pictures
Author Response
Dear Editor and Reviewers:
Thank you very much for your comments and suggestions.
Those comments are all valuable and very helpful for revising and improving our paper, as well as the important guiding significance to our researches. We have studied comments carefully and have made correction which we hope meet with approval. Revised portion are markedin red in the paper. The main corrections in the paper and the responds to the reviewer’s comments are as flowing:
Responds to the reviewer’s comments:
Reviewer 2:
1. Comment:
ABSTRACT:
L13: I don't think the word "gradient" is appropriate here. I will use word "treatment".
Response: Thank you very much for your valuable comments, I have replaced the word "gradient" with the word "treatment".
2.Comment:
MATERIAL AND METHODS:
In some places, the abbreviations which are used letter in the Results and Discussion sections in M&M are missing or there are abbreviations without explanation. Please revise through the M&M section.
L194-195: Why uppercase letters are used? The same for L198.
Response: Thank you very much for your valuable comments, I have checked the M&M Results and Discussion sections for missing or unexplained abbreviations used, and have made corrections. I have also removed capital letters.
3.Comment:
RESULTS:
For all tables in Results section please, arrange the text alignment.
Table 3: Please check the units.
Subtitle 3.2.1 is the same as 3.1.1.
There is no explanation which treatment represents foto A, B, and C in Figure 1.
L285: remove "and"
Figure 3B: Water saturation deficit is not mentioned in M&M section.
L342-346: The same statements are written twice. Please delete the first two sentences or delete the third one.
L360-364: Both sentences say very similar things. Please reformat the text to avoid such duplications. Do this throughout the Results chapter. In this way, you can greatly shorten a text that is relatively long.
Figure 8: explanation of which treatment represents foto A, B, and C is missing. The labels explained in the Notes below the figure are not visible in the pictures (very poor resolution).
Figure 10: very poor resolution for text in the pictures
Response: (1) Thank you very much for your valuable comments, I have text-aligned all tables in the results section.
- Thank you very much for your valuable comments, I have checked the table 3 units and it is correct.
- Thank you very much for your valuable comments, I have made changes to the subtitles part of 3.2.1 and 3.1.1. It is also explained in Figure 1 which process represents A, B and C. I have also deleted the redundant and on line 285.
- Thank you very much for your valuable comments, I have re-referenced the water saturation deficitin the M&M section.
- Thank you very much for your valuable comments, I have deleted similar sentences in the text.And I also have greatly shorten a text that is relatively long.
- Thank you very much for your valuable comments, I have explained below the figure which treatment represents A, B and C. And I also replaced the figure10.
- Best wishes,
Ying Liu
